# PAX6 protein in neuromasts of the lateral line system of salamanders (*Eurycea*)

**Brittany A. Dobbins**[1], **Ruben U. Tovar**[2], **Braden J. Oddo**[1], **Christian G. Teague**[1], **Nisa A. Sindhi**[1], **Thomas J. Devitt**[2], **David M. Hillis**[2], **Dana M. García**[1] *

1 Department of Biology, Texas State University, San Marcos, TX, United States of America, 2 Department of Integrative Biology, The University of Texas at Austin, Austin, TX, United States of America

* Dana_Garcia@txstate.edu

## Abstract

PAX6 is well known as a transcription factor that drives eye development in animals as widely divergent as flies and mammals. In addition to its localization in eyes, PAX6 expression has been reported in the central nervous system, the pancreas, testes, Merkel cells, nasal epithelium, developing cells of the inner ear, and embryonic submandibular salivary gland. Here we show that PAX6 also appears to be present in the mechanosensory neuromasts of the lateral line system in paedomorphic salamanders of the genus *Eurycea*. Using immunohistochemistry and confocal microscopy to examine a limited number of larvae of two species, listed by the United States of America's federal government as threatened (*E. nana*) or endangered (*E. rathbuni*), we found that anti-PAX6 antibody labeled structures that were extranuclear, and labeling was most intense in the apical appendages of the hair cells of the neuromast. This extranuclear localization raises the possibility of an as yet undescribed function for PAX6 as a cytoskeleton-associated protein.

**Data Availability Statement:** All relevant data are within the manuscript and its supporting information files. All confocal images from which counts of neuromasts with PAX6 labeling were

## 1. Introduction

PAX6 is well known as a transcription factor that drives eye development in animals as widely divergent as flies and mammals. In addition to its localization in eyes, including in limbal epithelial niche cells [1], PAX6 expression has been reported in the central nervous system, pancreas, and testes (see [2]), Merkel cells in whisker follicles [3], developing nasal epithelium [4], and developing cells of the inner ear [5], and the embryonic submandibular salivary gland [6]. In these tissues, PAX6 functions as a transcriptional regulator of cell proliferation and differentiation. When its expression is suppressed, as observed in Mexican cavefish (*Astyanax mexicanus*), eye development fails to progress: apoptosis leads to involution and the disappearance of the lens and an underdeveloped retina (reviewed in [7]).

Like the genus *Astyanax*, the salamander genus *Eurycea* includes aquatic species that live at the surface, species that live underground, and species that have populations in both environments (see [8], for a recent phylogeny). For example, the federally threatened San Marcos salamander (*E. nana*) lives in the surface waters emanating from the San Marcos Springs about 50 km south of Austin, Texas, USA. The federally endangered Texas blind salamander (*E.*

made can be found at https://doi.org/10.6084/m9.figshare.26240342.v1.

**Funding:** The equipment in the ARSC is supported by funds from the Materials Applications Research Center. This work was supported by the National Science Foundation [grant numbers DEB2032632 (DMH and TDH) and DEB203263 (DMG)]. The funders played no role in study design, data collection and analysis, decision to publish, or preparation of the manuscript.

**Competing interests:** The authors have declared that no competing interests exist.

*rathbuni*) inhabits the Edwards Aquifer in central Texas which periodically spews individuals to the surface via artesian wells in and around the city of San Marcos. The state threatened Cascade Caverns salamander (*E. latitans*) has both a surface and subterranean phenotype, the latter showing greatly reduced eyes along with several other characteristics typical of cave organisms.

As part of our work investigating *Pax6* expression during eye development in the surface-dwelling San Marcos salamander and the subterranean Texas blind salamander, we serendipitously observed labeling for PAX6 in the skin of both species. Here we present evidence that PAX6 is present in the mechanosensory neuromasts of the lateral line with particularly intense labeling in the apical appendages of the hair cells of the neuromast. This finding is surprising both because it is a rare instance of PAX6 labeling in the peripheral nervous system and because the labeling manifests extranuclearly, raising the possibility of an as yet undescribed function for PAX6.

## 2. Materials and methods

### 2.1 Animals

The protocols used in this research were reviewed and approved by The University of Texas at Austin, Institutional Animal Care and Use Committee (IACUC protocol approval number AUP-2021-00090). Salamanders were collected and transported under a Texas Parks and Wildlife Scientific Permit for Research (SPR-0119-004), and State Park permit number 57–22 issued to RUT. The San Marcos Aquatic Resources Center (SMARC) is operated by the United States Fish and Wildlife Service. It serves as a refugium for endangered species and maintains captive, breeding populations of San Marcos salamanders (*Eurycea nana*), Texas blind salamanders (*E. rathbuni*), as well as other threatened and endangered species (see https://www.fws.gov/office/san-marcos-aquatic-resources-center). Salamander larvae of *E. nana* and *E. rathbuni* (1 month post-oviposition, or MPO; Fig 1) along with an adult specimen of *E. rathbuni* were graciously provided by SMARC after euthanasia. Salamander larvae of *E. latitans* (1 MPO) were obtained from breeding populations at The University of Texas at Austin. Salamanders for immunolabeling were fixed on site in 4% formaldehyde (derived by alkaline depolymerization of paraformaldehyde) prepared in 0.1 M phosphate buffered saline (PBS), pH 7.4. Larvae for western blots were flash-frozen on-site using liquid nitrogen. Mouse eyes for western blot were generously donated by Texas State University's Dr. Ramona Price and flash frozen using liquid nitrogen.

### 2.2 Tissue preparation and sectioning

Twenty-four hours after specimens were fixed, they were washed three times in PBS for 15 minutes/wash and then incubated at 4˚C in 30% sucrose prepared in PBS with 0.1% sodium azide as an antimicrobial agent. Once specimens were cryoprotected, as indicated by sinking to the bottom of the tube, images of each individual were acquired using a Hirox digital microscope prior to processing for sectioning, and specimens were assigned unique identifying numbers and stored individually at 4˚C until use. Specimens were embedded in TissueTek® and cut into 15 μm sections at -22˚C using a Leica Cryostat equipped with a disposable blade. Sections were collected on gelatin-coated glass slides and stored at -20˚C until use.

### 2.3 Immunolabeling

Immunolabeling of tissue sections was accomplished as previously described [9] with minor alterations. In brief, sections of *E. nana* and *E. rathbuni* were incubated for one hour at room

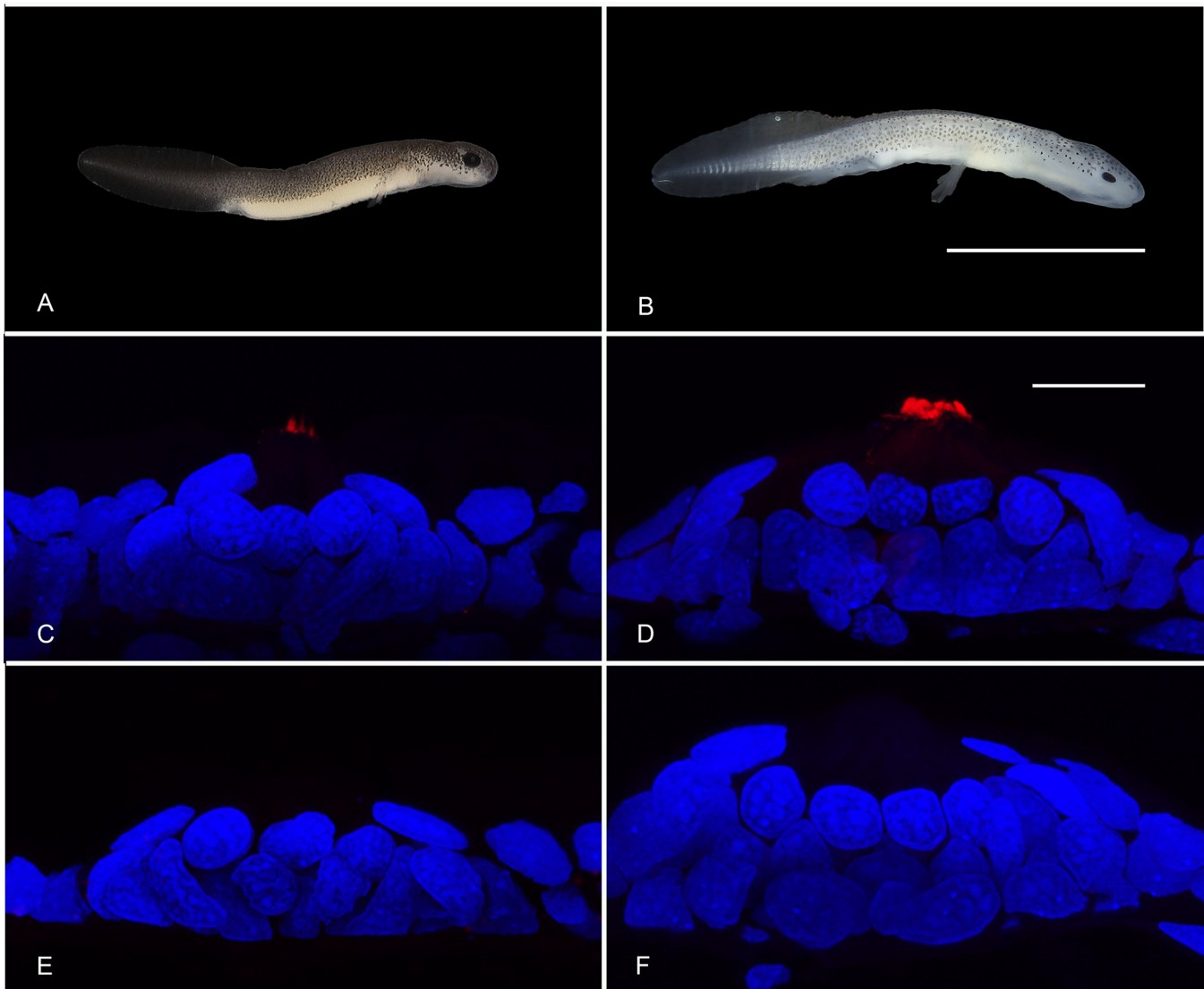

**Fig 1. PAX6 in hair cells of neuromasts of San Marcos salamander and Texas blind salamander larvae.** PAX6 (red) localizes to the apical appendages of hair cells of the neuromasts of the anterior lateral line system of both San Marcos salamander (A, C, E) and Texas blind salamander (B, D, F) larvae. Labeling appears in cilia-like projections originating at the center of the apical surface of the neuromasts (C, D). Negative control images (no exposure to anti-PAX6 antibody) for San Marcos (E) and Texas blind (F) salamanders are presented for comparison. The neuromasts shown are from the larvae shown in A and B. The scale bar in B is 5000 μm; the scale bar in D is 20 μm. Confocal images represent Z-projections acquired at the same magnification. Settings for detecting Cy5 (red) are the same for (C) and (D) with a laser power of 0.69%. The laser power for detecting Cy5 was 3.03% in (E) and (F), showing the lack of non-specific binding by the secondary antibody. Nuclei are labeled blue with DAPI.

temperature in PBS containing 0.3% Triton X-100 (LabChem, LC262801; PBST) and 5% non-fat dry milk as a blocking agent. Following blocking, sections were washed three times for 10 minutes/wash with PBS. Sections were then incubated with anti-PAX6 antibody (Thermo Fisher/Invitrogen PA1-801) diluted 1:75 in PBST containing 0.1% non-fat dry milk. Slides to serve as negative controls were either incubated with preadsorbed anti-PAX6 or not incubated with anti-PAX6, but with PBST containing 0.1% non-fat dry milk. Preadsorption was accomplished using custom peptide (Thermo Fisher) matching the antibody's immunogenic sequence at a concentration 5-times the molarity of the antibody. To maintain equal antibody concentration for both preadsorbed and experimental solutions, 0.1% non-fat dry milk in

PBST was added to the experimental solution at a volume equal to the peptide solution. Both solutions were incubated for 1.5 hours at room temperature with agitation, then applied to the slides. Following a two-hour incubation at room temperature, slides were washed three times for 10 minutes/wash in PBS. Sections were then incubated for one hour at room temperature with goat anti-rabbit secondary antibody conjugated to Cy5 (Thermo Fisher, A10523), diluted 1:500 in PBST containing 0.1% non-fat dry milk. Following this step, sections were washed three times for 10 minutes/wash in PBS. Coverslips were mounted onto the slides with Ever-brite™ Fluorescence Antifade Mounting Medium (Quartzy/Biotium, 23002) containing DAPI as a nuclear stain. Slides were stored frozen at -20˚C until used.

A western blot was performed to validate PAX6 antibody (Thermo Fisher/Invitrogen PA1-801) in *Eurycea*, using lysate of flash-frozen, adult mouse eye as a positive control [10] and pre-adsorbed antibody as a negative control. Due to the scarcity of specimens, larvae of *E. latitans*, a species closely related to *E. rathbuni* and within the same subgenus as *E. nana* (Eastern Blepsimolge; [8]) were used for the preadsorption analysis. Lysates were made using RIPA lysis buffer (Santa Cruz Biotechnology, sc-24948), protease inhibitor (a cocktail of pepstatin, leupeptin, AEBSF, E-64 and blebstatin kindly provided by Dr. Hong-Gu Kang at Texas State University), and a rotor-stator. Proteins were separated by SDS-PAGE using precast, Bis-Tris gradient (4% - 20%) gels (Genscript/Fisher, M00656) run at constant voltage. Molecular weight markers used were PageRuler Plus (prestained, 26619) or SuperSignal (unstained, 34580), designed to be visualized by chemiluminescent secondary antibodies. Following SDS-PAGE, proteins were transfered to nitrocellulose membrane which was subsequently cut in half to produce replicate membranes. Primary antibody was diluted 1:10,000 in 5% non-fat dry milk in Tris-buffered saline with Tween20 (TBST), and half was incubated with custom peptide (Thermo Fisher) matching the antibody's immunogenic sequence, resuspended in PBS at a concentration at least 5 times the molarity of the antibody. To maintain equal antibody concentration for both solutions, 5% non-fat dry milk in TBST was added to the other half of the primary antibody dilution at a volume equal to the peptide solution. Both solutions were incubated for 1.5 hours at room temperature with agitation. Membranes were blocked for 1 hour with 5% non-fat dry milk in TBST with agitation. One membrane was incubated with the peptide-bound antibody and the other with the unbound antibody overnight at 4˚ C. Membranes were washed three times for 10 minutes/wash with TBST and agitation, goat anti-rabbit HRP-linked secondary antibody was applied for 1 hour at room temperature with agitation. Membranes were washed three times for 10 minutes/wash with TBST and agitation. Chemiluminescent substrate (SuperSignal West Atto Ultimate Sensitivity Substrate, A38554) was used to visualize HRP labeling using an Azure c600 Gel Imaging System. To capture the location of the prestained PageRuler Plus molecular weight marker in the image generated by the gel imaging system, two triangular notches were cut from the membrane next to the ladder pointing to 30 kDa and 50 kDa.

## 2.4 Image acquisition and preparation

Images of immunolabeled sections were obtained using an Olympus FV1000 laser scanning confocal microscope. A minimum of three neuromasts from each of three individuals of each of the two species were imaged using a 60x oil immersion objective with a numerical aperture of 1.42. Confocal laser settings were optimized for the DAPI channel for each section. The laser settings for the Cy5 channel were optimized using a sample generated from a Texas blind salamander, and the same Cy5-settings were used for all images acquired. Images were captured as Z-stacks comprising 30 optical sections with a step size of 0.41 μm and were prepared for publication using Adobe Photoshop 24.5.0.

## 2.5 Sequence analysis

Using a comprehensive list of human PAX protein sequences and NCBI accession numbers provided by Thompson et al. [2] as a starting point, sequences of all PAX proteins were downloaded and aligned using Geneious Prime 2023.1.2. Species that shared taxonomic similarities with *Eurycea* salamanders which had PAX6 sequences were obtained as well and aligned against the human PAX6 sequences. The PAX6 sequence for axolotl (*Ambystoma mexicanum*) was acquired from a BLAST search of the axolotl proteome. We acquired the Iberian ribbed newt (*Pleurodeles waltl*) sequence by BLAST using the immunogenic sequence and observing the top 100 results that exhibited the immunogenic sequence. The PAX6 sequence for Japanese fire-bellied newt (*Cynops pyrrhogaster*) was also uncovered using BLAST with results filtered by the superfamily Salamandroidea and having a percent identity of 95%-100%. These sequences were aligned, and a figure was generated using Geneious. Sequence data are not publicly available for salamanders in the genus *Eurycea*.

RNA was isolated from different tissues of an adult *Eurycea nana* and sequenced using an Illumina HighSeq (2x250 base pair, paired-end). A *de novo* transcriptome was then assembled using Trinity. Sequences were mapped to an annotated transcriptome from *E. paludicola*. A partial sequence of *E. nana* PAX6 was added to the alignment figure in Adobe Photoshop.

## 3. Results

Immunolabeling with anti-PAX6 antibodies revealed labeling of the cells of the neuromasts of the San Marcos salamander and the Texas blind salamander. The labeling was particularly intense in the apical appendages of the hair cells (Fig 1). This pattern of labeling was evident at low laser power (0.69%) in five of the fourteen neuromasts observed in the Texas blind salamander and eight out of thirteen observed at high laser power (3.03%). Five neuromasts were imaged at both laser settings; one showed labeling in the apical appendages only at the higher laser setting, and two showed labeling at both the higher and lower setting. At the individual level, two out of three larvae exhibited labeling in the apical appendages at low laser power, and all three showed labeling at high laser power. Ten of the ten neuromasts observed in the San Marcos salamander showed labeling at low laser power. A different subset was imaged at high laser power, and ten out of ten showed labeling.

At higher laser power, fluorescence was also observed in the perinuclear cytoplasm of hair cells; however, at the lower laser power labeling of the perinuclear cytoplasm was absent even in cells in which labeling of the apical appendages was still apparent. At lower laser power, labeling within the nucleus of neuromast cells was not observed, nor was labeling in the other cells of the neuromasts or surrounding epithelium (Figs 1 and S1). PAX6-labeling was also observed in the apical appendages of hair cells in the neuromasts of adult, Texas blind salamander (S2 Fig).

To test whether the antibody was recognizing salamander PAX6 in tissue sections, we preabsorbed the antibody with a custom-made peptide comprising the immunogenic sequence published by the manufacturer. Preadsorption of the primary antibody eliminated labeling in the apical processes of the hair cells (S3 Fig). Additionally, western analysis indicated that the primary antibody binds a protein of about 36 kDa in both mouse eye homogenates and salamander lysates (S4 Fig). Binding to the 36 kDa protein was eliminated by preabsorbing the antibody with its immunogen (S4 Fig).

To further address the question of whether the anti-PAX6 antibody (raised against an immunogenic sequence corresponding to mouse PAX6 and identical to human PAX6) would recognize all the isoforms of PAX6, we analyzed the sequences of all human PAX6 proteins. This analysis revealed that all fifteen isoforms of human PAX6 contain the immunogenic

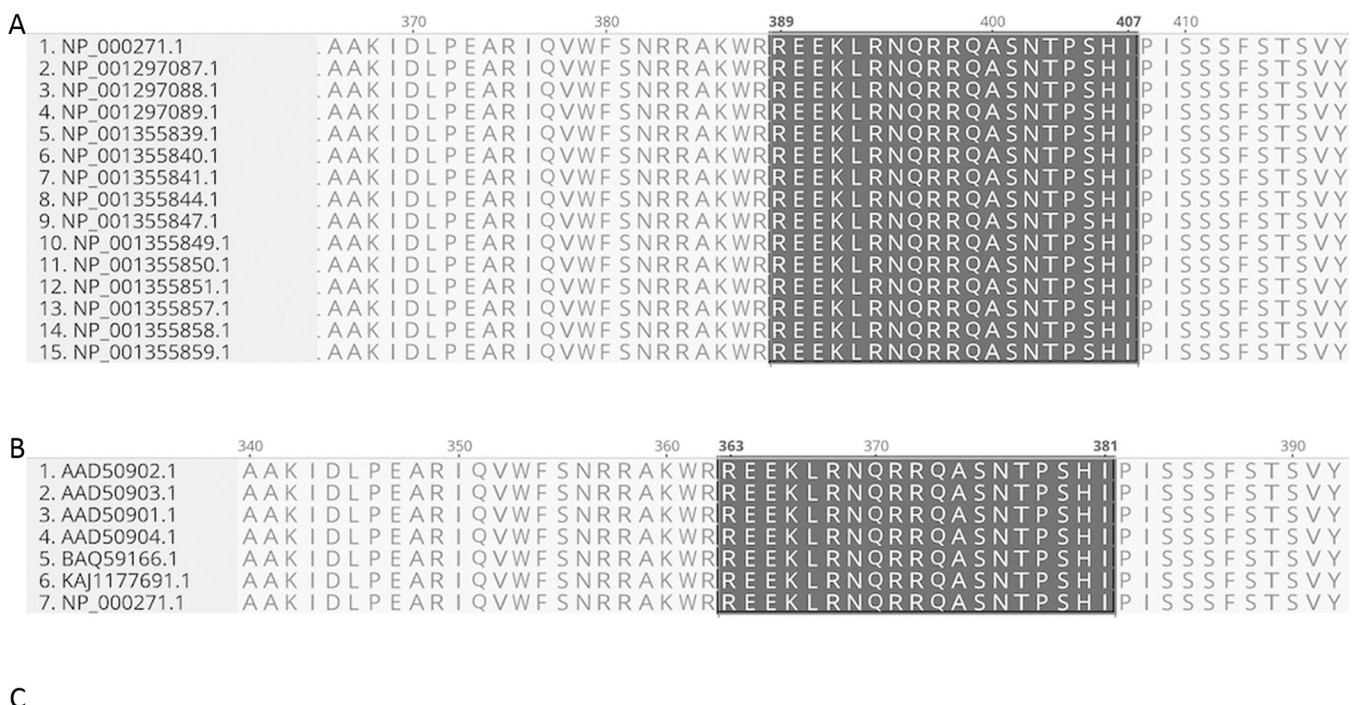

**Fig 2. Sequence analysis of PAX6 proteins.** (A) An alignment of human PAX6 sequences reveals that all fifteen isoforms share the immunogenic sequence (highlighted in dark grey). NCBI sequence numbers are to the left of the sequences which are arranged alphabetically by isoform. (B) The four isoforms of axolotl PAX6 (sequences 1–4) contain the immunogenic sequence as does PAX6 from the Japanese fire-bellied newt (sequence 5) and the hypothetical protein from the Iberian ribbed newt (sequence 6). Sequence 7 is human PAX6. (C) Partial sequence obtained from *E. nana* containing 16/19 amino acids from the PAX6 immunogenic sequence.

sequence published by ThermoFisher, specifically REEKLRNQRRQASNTPSHI (Fig 2A). Additionally, none of the other human PAX proteins contained that sequence. A search of the NCBI database using BLAST and the immunogenic sequence as the query sequence yielded 100 sequences containing the identical sequence, 95 of which were identified explicitly as PAX6 sequences. Those sequences that were not explicitly labeled as PAX6 sequences were hypothetical proteins inferred from genome projects and containing PAX6-like sequences, including from the Iberian ribbed newt (*Pleurodeles walti*) and the Japanese fire-bellied newt (*Cynops pyrrhogaster*), both species in the same superfamily as the San Marcos and Texas blind salamanders, namely Salamandroidea [11]. PAX6 sequences for axolotl (*Ambystoma mexicanus*), another member of the Salamandroidea [11], also revealed 100% identity with the immunogenic sequence. Comparing these sequences showed an identical region of 19 residues making up the immunogenic sequence (Fig 2B), and 92.4% identity overall by pairwise comparison. Additional differences include variability in the length of the predicted sequences; for example, in the Iberian ribbed newt, the predicted sequence has an additional 81 residues at the N-terminus. Sequence inferred from the *Eurycea nana* transcriptome included 16 amino acids of the immunogenic sequence, with the three N-terminal amino acids missing from the inferred amino acid sequence, probably due to truncation of the mRNA in the transcriptomic analysis (Fig 2C). Consistent with our sequence analysis, western analysis indicated that the antibody recognized a protein in *E. nana*, *E. rathbuni*, and *E. latitans* that co-migrated with a protein derived from mouse retina (see S4 Fig).

## 4. Discussion

We explored the localization of PAX6 in larvae of the San Marcos salamander and the Texas blind salamander. We found expression in the eye as expected based on previous observations of Texas blind salamander and Barton Springs salamander (*Eurycea sosorum*) [9] and unexpectedly in the neuromasts of the anterior lateral line system. PAX6 localization in the nervous system has been well studied with a particular focus on the central nervous system where its nuclear localization befits its role as a transcription factor. Parisi and Collinson [3] observed PAX6 in Merkel cells, mechanoreceptors associated with whisker follicles, and they reported its localization in developing mice as young as E16.5. Like us, they were surprised to see PAX6 localized to the cytoplasm of Merkel cells of E18.5 mice but noted that at post-natal day 4, localization appeared to be shifting toward the nucleus. While they did not show results from examining skin from older individuals, previous work from their lab showed cytoplasmic localization of Pax6 associated with oxidative-stress of corneal cells from adult *Pax6*$^{+/-}$ mice [12]. Similarly, Parisi and Collinson [3] found that in cultured Merkel cells, they could both simulate the transition of PAX6 labeling from cytoplasmic to nuclear over time and induce nucleocytoplasmic shuttling by exposing the cells to $H_2O_2$ to induce oxidative stress. In contrast, we observe that the localization of PAX6 in the apical appendages of the hair cells occurs under ostensibly normal conditions of development and persists in adults, at least in the case of the Texas blind salamander.

The nucleocytoplasmic shuttling of transcription factors observed by Parisi and Collinson [3] has also been reported in other cases, where it may play a role in enabling or disabling these factors from influencing transcriptional activity. For example, cytoplasmic steroid hormone receptors bind their cognate hormone and then translocate to the nucleus where they serve as transcription factors, modulating the expression of target genes in order to transduce hormonal signals [13]. Their transcriptional efficiency necessarily depends then on their subcellular distribution. Throughout life, steroid receptors are continually transported between the nucleus and cytoplasm, and this trafficking can represent a nonpathological role for nucleocytoplasmic shuttling (stress-induced steroid release notwithstanding).

Another example of a transcription factor that can be found distributed between the nucleus and cytoplasm is heat shock factors (HSFs), which act as transcription factors safeguarding cells against various forms of stress by regulating heat shock genes responsible for producing heat shock proteins [14]. In contrast to the developmental, cytoplasmic to nuclear movement reported for PAX6 in Merkel cells [3], Brown and Rush [14] reported translocation of HSF2 from the nucleus (post-natal day 2) to the cytoplasm (post-natal day 30) in rat brain, ostensibly as a normal part of development. Interestingly cytoplasmic localization of HSF2 persisted even after cells were subjected to a fever-like temperature shock. Similarly, HSF1's localization to the nucleus seemed unaffected by elevated temperature. These observations along with others dissociating HSF2 localization from production of heat shock proteins [see 14, "Discussion"], led Brown and Rush to propose that HSF2 was involved in other, non-transcriptional processes. Similarly, our observation that PAX6 localizes to the apical appendages of hair cells leads us to hypothesize that PAX6 has an as yet uncharacterized function beyond transcriptional regulation.

Alastalo et al. [15] also observed a shift of HSF2 from nucleus to cytoplasm during spermatogenesis, and they proposed HSF2 was involved in cellular processes in addition to its role as a transcription factor. They demonstrated that HSF2 is present in the nuclei of early pachytene spermatocytes at stages I-IV and in the nuclei of round spermatids at stages I-VIIab. In addition, strong HSF2 immunoreactivity was detected in small, distinct cytoplasmic regions from zygotene spermatocytes to maturation-phase spermatids. Additionally, they presented

immunoelectron microscopic analysis showing HSF2 localizing to cytoplasmic bridges between germ cells and proposed HSF2 played a role in enabling gene products to be shared between cells.

The pattern of PAX6 labeling we observed in neuromast is striking and unique, as it is strongly concentrated in the ciliary region of hair cells of neuromasts. Like Merkel cells [16], hair cells of the neuromasts are ectodermally derived mechanoreceptors equipped to synapse on afferent neurons of the somatosensory system (see [17] for review). The hair cells are so called because of the stereocilia and kinocilia that extend from their apical surfaces supported by actin filaments and microtubules, respectively, to form the hair bundle (see [18]). The labeling we observed in the neuromasts was most intense in the apical appendages, which we presume to be the stereocilia because of their abundance. Studies are underway to determine whether PAX6 associates with specific cytoskeletal elements in these structures.

The PAX6 labeling that we observed in apical appendages is evident in neuromasts of adult Texas blind salamanders (S2 Fig) as well as larval salamanders. Interestingly, these salamanders share a paedomorphic life history [19], maintaining larval features throughout their lives. At the tissue level, paedomorphy is thought to occur by slowing the developmental rate of somatic tissue while maintaining gonadal tissue development and eventual maturation, a condition known as neoteny [20]. Life history may play a role in the progressive loss or gain of neuromasts. Loss of lateral line neuromasts is observed in some terrestrial salamander species that undergo direct development, i.e., omitting the larval stage. During development of the lateral line, the afferent lateral line neurons innervate the skin prior to neuromast formation [21]. Interestingly, the underlying afferent lateral line neurons can still be observed within some species of dusky salamanders (genus *Desmognathus*), for example, seepage salamanders (*D. aeneus*) and pygmy salamanders (*D. wrighti*) in the absence of neuromasts [22]. The retention of afferent lateral line neurons has been interpreted as neuro-anatomical paedomorphy [23,24]. It would be interesting to explore whether neuromasts of metamorphosing salamanders that maintain a lateral line similarly evince PAX6 in their neuromasts.

A challenge of extending the work we present here is the availability of specimens of these understudied, threatened and endangered species for which publicly available genomic data is not available. Work is underway to generate transcriptomic data, and preliminary results for the sequence of *Pax6* from the transcriptome of *Eurycea nana* are included here. We expect to further investigate *Pax6* expression in skin, and more precisely in neuromasts in future studies, contingent on the availability of salamander larvae. The immuno-histochemical results we report here are nevertheless provocative, and though the number of individuals probed with anti-PAX6 antibodies is small, it is worth noting that labeling of the apical appendages of hair cells in the neuromasts was observed in all of them. That it was not observed in all neuromasts could be attributed to a failure to capture the apical appendages in some tissue sections.

## 5. Conclusions

We have shown here that anti-PAX6 antibodies label neuromasts of the San Marcos salamander and the Texas blind salamander, localizing in the apical appendages of the hair cells. Sequence analysis supports our inference that the antibody recognizes PAX6 and not a similar protein. This inference is corroborated by experiments in which preadsorption of antibodies with the immunogenic peptide eliminates labeling in both tissue sections and westerns. Future studies will be directed toward better resolving the precise association of PAX6 protein with structures in the hair cells.

## Supporting information

**S1 Fig. PAX6 labeling in neuromasts is evident only where hair cells have been cross sectioned.** An arrowhead points to the PAX6 labeling (red) of apical projections of hair cells in the cross sectioned neuromast. The neuromast to the left, which was sectioned off-center (evident by the mantle cells extending over the apical surface), does not show PAX6 labeling. The scale bar in the image is 100 µm. Image represents Z-projections acquired using a 20x air objective with a numerical aperture of 0.75.
(TIF)

**S2 Fig. PAX6 protein localizes to the apical appendages of hair cells in neuromasts of adult Texas blind salamanders.** PAX6 (red) localizes to the apical appendages (arrowhead) of hair cells in the neuromast of an adult Texas blind salamander. The scale bar in the image is 20 µm, and the image represents a Z-projection acquired at the same magnification and settings as used in Fig 1(E,F). Nuclei are labeled blue with DAPI.
(TIF)

**S3 Fig. Antibody validation using preabsorbed antibodies in immunohistochemistry.** IHC images of *E. rathbuni* experimental (A) and negative control (B,C) slides. The PAX6 labeling in neuromasts of the experimental sections (A) is not detected in preadsorbed primary antibody sections (C) or the no primary antibody sections (D) serving as negative controls. Arrowheads point towards the apical appendages of hair cells in neuromasts. The scale bars in the images are 20 µm, and the images represent Z-projections acquired at the same magnification and settings. Nuclei are labeled blue with DAPI.
(TIF)

**S4 Fig. Antibody validation using western analysis.** Tissue lysates were obtained from embryos of *E. rathbuni*, *E. nana*, and *E. latitans* as well as adult mouse eye to serve as a positive control. PagerRuler Plus (Thermo Fisher, cat #26619) [A] and SuperSignal (Thermo Fisher, cat # 84785) molecular weight markers [B,C] were used to estimate the molecular weight of labeled bands. A western analysis using our primary PAX6 antibody suggests that the antibody labels a protein of about 36 kDa in all four taxa. (A) Arrowheads are overlaid on each triangular notch cut into the membrane. *E. rathbuni*, *E. nana*, and mouse eye lysates have one labeled band in common at around 36 kDa. A broad, labeled region can be seen in the mouse eye lane between 20 kDa and 27 kDa. (B) *E. latitans* and mouse eye have one labelled band in common at around 36 kDa. The dotted line indicates where replicate lanes were cropped from the image. The lanes to the right of the dotted line were flipped horizontally to match the lane order of those to the left. (C) Preadsorbed antibody does not bind to the 36 kDa protein from *E. latitans* or mouse eye. Labeling of the lower molecular weight bands in mouse eye is also absent following preadsorption of the primary antibody.
(TIF)

**S5 Fig. Raw western blot images.** Raw images used to construct S4 Fig. "X" signifies lanes that were cropped in the construction of S4 Fig, in which their former location is indicated by a vertical, dashed line. The image in [A] is inverted in S4A. Western blots were prepared as follows: Tissue lysates were obtained from embryos of *E. rathbuni*, *E. nana*, and *E. latitans* as well as adult mouse eye to serve as a positive control. PagerRuler Plus (Thermo Fisher, cat #26619) [A] and SuperSignal (Thermo Fisher, cat # 84785) molecular weight markers [B,C] were used to estimate the molecular weight of labeled bands. A western analysis using our primary PAX6 antibody suggests that the antibody labels a protein of about 36 kDa in all four taxa. (A) Arrowheads are overlaid on each triangular notch cut into the membrane. *E. rathbuni*, *E.*

*nana*, and mouse eye lysates have one labeled band in common at around 36 kDa. A broad, labeled region can be seen in the mouse eye lane between 20 kDa and 27 kDa. (B) At the highest lysate concentrations in a dilution series, *E. latitans* and mouse eye have one labelled band in common at around 36 kDa. (C) Preadsorbed antibody does not bind to the 36 kDa protein from *E. latitans* or mouse eye. Labeling of the lower molecular weight bands in mouse eye is also absent following preadsorption of the primary antibody.
(TIF)

## Acknowledgments

We are grateful to the San Marcos Aquatic Resources Center (United States Fish and Wildlife Service) for donation of salamander larvae. The staff at the Applied Research Service Center (ARSC) of Texas State University, especially Mrs. Alissa Savage, Mr. Jacob Bisbal and Mr. Jacob Armitage, made every accommodation possible to assure BAD was trained efficiently on the confocal and digital microscopes during her tenure as an NSF Summer Research Fellow. Mrs. Savage continues to provide helpful, technical assistance. Dr. Hong-Gu Kang of Texas State University provided protease inhibitors and generously allowed access to his laboratory's gel imaging system. Dr. Ramona Price of Texas State University kindly donated mouse tissue.

## Author Contributions

**Conceptualization:** Ruben U. Tovar.

**Data curation:** Thomas J. Devitt.

**Funding acquisition:** Thomas J. Devitt, David M. Hillis, Dana M. García.

**Investigation:** Brittany A. Dobbins, Braden J. Oddo, Christian G. Teague.

**Project administration:** Thomas J. Devitt, David M. Hillis, Dana M. García.

**Resources:** Thomas J. Devitt.

**Supervision:** Ruben U. Tovar, Thomas J. Devitt, Dana M. García.

**Visualization:** Brittany A. Dobbins, Braden J. Oddo, Dana M. García.

**Writing – original draft:** Dana M. García.

**Writing – review & editing:** Brittany A. Dobbins, Ruben U. Tovar, Braden J. Oddo, Nisa A. Sindhi, Thomas J. Devitt, David M. Hillis.

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
