## [Decision Letter · Decision Letter 0]

28 Dec 2023

PONE-D-23-32484PAX6 protein in neuromasts of the lateral line system of salamanders (*Eurycea*)PLOS ONE

Dear Dr. Garcia,

Thank you for submitting your manuscript to PLOS ONE. After careful consideration, we feel that it has merit but does not fully meet PLOS ONE’s publication criteria as it currently stands. Therefore, we invite you to submit a revised version of the manuscript that addresses the points raised during the review process. Both reviewers have serious concerns about the specificity of the Pax6 expression data and its developmental implications.  It seems clear that in situ hybridization data is needed to more rigorously establish the pattern of Pax6 expression in neuromasts, and that at least a attempt to characterize Pax6 expression during development (not at a single stage) should be presented, so that the behavior in this species can be better related to previous work.  An alternative to in situ hybridization would be to absorb the antiPax6 antibody with the antigen against which it was raised (Pax6 protein?).  A point by point response to the reviewers comments should be presented in a revised manuscript.  Also, the figure legends should be "broken out" of the main text (I had a little trouble finding them).  

We look forward to receiving your revised manuscript.

Kind regards,

Michael Klymkowsky, Ph.D.

Academic Editor

PLOS ONE

Reviewers' comments:

Reviewer's Responses to Questions

**Comments to the Author**

1. Is the manuscript technically sound, and do the data support the conclusions?

Reviewer #1: Partly

Reviewer #2: No

2. Has the statistical analysis been performed appropriately and rigorously? 

Reviewer #1: N/A

Reviewer #2: N/A

3. Have the authors made all data underlying the findings in their manuscript fully available?

Reviewer #1: Yes

Reviewer #2: Yes

4. Is the manuscript presented in an intelligible fashion and written in standard English?

Reviewer #1: Yes

Reviewer #2: No

5. Review Comments to the Author

Reviewer #1: This is a short research note that documents an interesting observation that the Pax6 transcription factor protein appears (at level of immunohistochemistry) to localise to neuromasts in two species of paedomorphic salamanders in the genus Eurycea. The protein is localised to the cytoplasm/cytoskeletal elements of the cells rather than the nucleus, suggesting a role different from gene regulation.

The authors have gone to some trouble to show, through no-primary-antibody controls, western blot and sequence homology comparisons, that the antibody labelling is likely to be specific for Pax6 and not another Pax protein.

I have the following comments. Essentially, while the observation is clearly presented, the data are rather thin and selective:

1) Please provide a more convincing and properly described western blot in FigS1. Needs size markers and a better image. This doesn't look like a western.

2) Weak cytoplasmic labelling in cells surrounding the neuromasts is observed - please clarify/show whether this persists in skin epidermis away from the neuromasts.

3) A much better and more convincing control would be to preabsorb/mix the antibody with the peptide against which it was raised, before addition to tissues. Without this and/or another assay (e.g. in situ or an antibody that recognises a different Pax6 epitope) to show expression in and around neuromasts, there remains some doubt as to whether the staining pattern is real.

4) Cytoplasmic localisation of transcription factors proteins is a widespread and well documented phenomenon and cytoplasmic/nuclear shuffling is a mainstream method of activity control (not just for Pax6). This should be discussed.

5) The authors should possibly also mention Pax6 expression in vertebrate salivary glands, which from figures also appears to be cytoplasmic e.g. https://karger.com/cto/article/170/2-3/83/90290/Embryonic-Submandibular-Gland-Morphogenesis-Stage

6) Line 28. The limbal epithelial niche is also the eye, not 'in addition' to the eye.

Reviewer #2: The authors report an observation that Pax6 may be expressed in the cytoplasm of lateral line neuromasts in San Marcos and Texas blind salamander larvae. After reading the manuscript, I am not fully convinced by the specificity of the staining. In addition, the study is very superficial and falls short of providing a relatively detailed description of Pax6 expression in neuromasts.

1. The specificity of Pax6 staining in neuromasts needs to be more rigorously controlled. In addition to omitting the primary antibody, non-immune serum or isotype control should be used in immunofluorescence experiments.

2. In situ hybridization should be performed to confirm Pax6 expression in neuromasts.

3. The authors did not explain in detail the localization of Pax6 in neuromasts. In particular, how it is localized in hair cells versus supporting cells.

4. The study is very superficial. The authors only examined Pax6 expression in neuromasts at one stage of development. They should perform immunostaining at different stages of neuromast formation.

5. Figure 2 is problematic. No San Marcos and Texas blind salamander Pax6 sequences are included in the alignment. It is unclear whether Marcos and Texas blind salamander Pax6 also contain the same immunogenic sequence.

6. There is no protein markers in Figure S2, and size of proteins bands is not clear. In addition, an antibody that can be used for western blot does not necessarily work in immunofluorescence. The authors also do not explain how the labeled protein bands are visualized.

6. PLOS authors have the option to publish the peer review history of their article (what does this mean?). If published, this will include your full peer review and any attached files.

Reviewer #1: **Yes: **Jon Martin Collinson

Reviewer #2: No

---

## [Author Response · Author response to Decision Letter 0]

27 May 2024

We are grateful for the constructive comments and advise from the reviewers. Please find our responses below:

Response to editor’s summary: 

Although our research program overall addresses questions relating to evolutionary divergence and the molecular changes that lead to different phenotypic outcomes, the main point of this paper is the appearance of PAX6 in the apical appendages of the hair cells of the mechanosensory neuromasts. Naturally, we are interested in following PAX6 localization during development; this question is addressed in a separate manuscript currently in preparation. 

Given the long evolutionary divergence time between amphibians and mammals, we also were concerned about whether the anti-PAX6 antibody raised against mouse PAX6 would recognize salamander PAX6. The salamanders we study are not model organisms; their transcripts and gene sequences are not currently represented in a publicly accessible database. At the time the original manuscript was submitted, we did not have any sequence data for Pax6 from either the San Marcos salamander (Eurycea nana) or the Texas blind salamander (E. rathbuni). We now have a partial sequence for E. nana; however, we do not yet have probes for in situ hybridization. We have, however, been able to do the immunolabeling experiments using antibodies pre-adsorbed with the immunogenic sequence published by the manufacturer as suggested. Furthermore, we have run new westerns, also using preabsorbed antibodies. Because embryos for the endangered species originally under investigation were not available, we used a related species (the Cascade Caverns salamander, E. latitans) for these experiments. The new micrographs and blots can be found in the Supporting Information.

Response to the reviewers’ comments (copied from e-mail with our responses bulleted)

Reviewer #1: This is a short research note that documents an interesting observation that the Pax6 transcription factor protein appears (at level of immunohistochemistry) to localise to neuromasts in two species of paedomorphic salamanders in the genus Eurycea. The protein is localised to the cytoplasm/cytoskeletal elements of the cells rather than the nucleus, suggesting a role different from gene regulation.

The authors have gone to some trouble to show, through no-primary-antibody controls, western blot and sequence homology comparisons, that the antibody labelling is likely to be specific for Pax6 and not another Pax protein.

I have the following comments. Essentially, while the observation is clearly presented, the data are rather thin and selective:

1) Please provide a more convincing and properly described western blot in FigS1. Needs size markers and a better image. This doesn't look like a western.

• The image of the original western has been reformatted to better represent its appearance. Additional westerns have also been included to address the question of specificity of the antibody by pre-adsorbing the primary antibody prior to its application to control westerns. While the original western used standard protein molecular weight markers (invisible to our imaging system, necessitating notching the membrane to mark the location of the 50 kDa and 30 kDa bands), the new westerns employ molecular weight markers designed for use with chemiluminescence. Both are now presented in S4 Fig. 

• A description of the methods by which the westerns were obtained has been added to “Materials and Methods.”

2) Weak cytoplasmic labelling in cells surrounding the neuromasts is observed - please clarify/show whether this persists in skin epidermis away from the neuromasts.

• For the revised manuscript, we re-imaged the larvae using lower laser settings. Our earlier images were prepared using somewhat high (3%) laser power, and we think that may have given the false impression of more widespread labeling than may be biologically relevant. Labeling persists in the apical appendages of hair cells in the new images acquired at 0.67% laser power; however, the weak cytoplasmic labeling in cells surrounding the neuromasts is not detected. A low magnification image (S1 Fig) including a labeled neuromast was obtained using a 20X (0.75 NA) lens and shows more of the surrounding tissue. 

3) A much better and more convincing control would be to preabsorb/mix the antibody with the peptide against which it was raised, before addition to tissues. Without this and/or another assay (e.g. in situ or an antibody that recognises a different Pax6 epitope) to show expression in and around neuromasts, there remains some doubt as to whether the staining pattern is real.

• The requested controls have been done and, as mentioned above, extended to Western analyses. Preabsorbing the antibody with the peptide published by the manufacturer as the immunogen eliminated labeling in the apical processes and cytoplasm (S3 Fig B). 

• In the western analysis, preabsorption eliminated labeling of co-migrating bands at 36 kDa in tissue homogenates obtained from Eurycea latitans and mouse eye. It also eliminated lower molecular weight bands observed in the mouse eye homogenate. A faint band migrating at 100 kDa persists in the E. latitans homogenate. Still, because the immunolabeling evident by confocal disappears, we do not think this band accounts for the labeling apparent in the apical appendages.

4) Cytoplasmic localisation of transcription factors proteins is a widespread and well documented phenomenon and cytoplasmic/nuclear shuffling is a mainstream method of activity control (not just for Pax6). This should be discussed.

• We have added to the discussion examples of transcription factors (steroid hormone receptors and heat shock factor 2) translocating between the cytoplasm and nucleus. The example of HSF2 was interesting because of its postulated role in forming cytoplasmic bridges between cells of the testis. 

5) The authors should possibly also mention Pax6 expression in vertebrate salivary glands, which from figures also appears to be cytoplasmic e.g. https://karger.com/cto/article/170/2-3/83/90290/Embryonic-Submandibular-Gland-Morphogenesis-Stage

• We thank the reviewer for directing our attention to this paper. We now mention this paper in our introduction as follows:

In addition to its localization in eyes, including in limbal epithelial niche cells [1], PAX6 expression has been reported in the central nervous system, pancreas, and testes (see [2]), Merkel cells in whisker follicles [3], developing nasal epithelium [4], developing cells of the inner ear [5], and the embryonic submandibular salivary gland [6]. 

6) Line 28. The limbal epithelial niche is also the eye, not 'in addition' to the eye.

• We restructured the sentence to convey this fact. Please see above.

Reviewer #2: The authors report an observation that Pax6 may be expressed in the cytoplasm of lateral line neuromasts in San Marcos and Texas blind salamander larvae. After reading the manuscript, I am not fully convinced by the specificity of the staining. In addition, the study is very superficial and falls short of providing a relatively detailed description of Pax6 expression in neuromasts.

1. The specificity of Pax6 staining in neuromasts needs to be more rigorously controlled. In addition to omitting the primary antibody, non-immune serum or isotype control should be used in immunofluorescence experiments.

• Please see the above response to the editor’s summary and Reviewer 1’s comment 3. 

2. In situ hybridization should be performed to confirm Pax6 expression in neuromasts.

• Please see the above response to the editor’s summary

3. The authors did not explain in detail the localization of Pax6 in neuromasts. In particular, how it is localized in hair cells versus supporting cells.

• We have reimaged the samples, and now include the following text to explain the localization of the PAX6-labeling:

Immunolabeling with anti-PAX6 antibodies revealed labeling of the neuromasts of the San Marcos salamander and the Texas blind salamander. The labeling was particularly intense in the apical appendages of these cells (Fig. 1). This pattern of labeling was evident in five of the ten neuromasts observed in the Texas blind salamander and ten of the eleven neuromasts observed in the San Marcos salamander. At higher laser power (3.03%), fluorescence was also observed in the perinuclear cytoplasm of hair cells; however, at the lower laser power (0.69%) that still revealed apical appendages, labeling of the perinuclear cytoplasm was faint. Three of ten hair cells from the Texas blind salamander showed labeling in the perinuclear cytoplasm; none of the hair cells from the San Marcos salamander showed perinuclear labeling. Labeling within the nucleus of neuromast cells was not observed, nor was labeling in the other cells of the neuromasts or surrounding epithelium (Fig 1 and S1 Fig). PAX6-labeling was also observed in the apical appendages of hair cells in the neuromasts of adult, Texas blind salamander (S2 Fig).

4. The study is very superficial. The authors only examined Pax6 expression in neuromasts at one stage of development. They should perform immunostaining at different stages of neuromast formation.

• As mentioned in the response to the editor’s summary above, we are interested in questions of the evolutionary divergence of developmental processes that lead to different phenotypes; however, in this paper we sought to highlight an interesting aspect of cell biology - the novel localization of PAX6 in association with apical appendages of hair cells in developing as well as adult salamanders - at least in the case of the adult Texas blind salamander. Our hope is that this finding will pique the interest of other scientists interested in cytoskeleton-associated proteins and other questions of cell and neurobiology. We have not yet examined early-stage embryos to test whether PAX6 can be found in neuromasts at earlier stages of development. We do not currently have access to earlier stages; however, we think the question of when PAX6 appears in the hair cells is an interesting one, and one which we hope to pursue.

5. Figure 2 is problematic. No San Marcos and Texas blind salamander Pax6 sequences are included in the alignment. It is unclear whether Marcos and Texas blind salamander Pax6 also contain the same immunogenic sequence.

• When the manuscript was submitted, no sequence data were available for these salamander species. During revision, we have gained access to a partial sequence for the San Marcos salamander; however, we still do not have a sequence for the Texas blind salamander. The partial sequence inferred from transcriptomic data is now included in the revised Fig 2, and the methodology for acquiring the sequence in included in the Materials and Methods section.

6. There is no protein markers in Figure S2, and size of proteins bands is not clear. In addition, an antibody that can be used for western blot does not necessarily work in immunofluorescence. The authors also do not explain how the labeled protein bands are visualized.

• The western blots are now in a revised S4 Fig, and the methods for obtaining them are now described. In brief, the molecular weight markers in the original western were prestained and visible to the naked eye, but not to the imaging system used to detect chemiluminescence. Therefore, the nitrocellulose membranes were marked by cutting, and in the revised figure, triangles are placed where the cuts were.

• For the preadsorption study also included in the revised figure, molecular weight markers adaptable to that purpose were used and can be seen directly in the figure.

We are grateful for the constructive criticism made by the editor and reviewers, and we hope that you will find the revised manuscript, though still brief, much improved by your input.

---

## [Decision Letter · Decision Letter 1]

11 Jun 2024

PONE-D-23-32484R1PAX6 protein in neuromasts of the lateral line system of salamanders (*Eurycea*)PLOS ONE

Dear Dr. Garcia,

Thank you for submitting your manuscript to PLOS ONE. After careful consideration, we feel that it has merit but does not fully meet PLOS ONE’s publication criteria as it currently stands. Therefore, we invite you to submit a revised version of the manuscript that addresses the points raised during the review process.

Please explicitly address reviewer #2's comments - I do not think new studies are necessarily needed. 

We look forward to receiving your revised manuscript.

Kind regards,

Michael Klymkowsky, Ph.D.

Academic Editor

PLOS ONE

Journal Requirements:

Reviewers' comments:

Reviewer's Responses to Questions

**Comments to the Author**

1. If the authors have adequately addressed your comments raised in a previous round of review and you feel that this manuscript is now acceptable for publication, you may indicate that here to bypass the “Comments to the Author” section, enter your conflict of interest statement in the “Confidential to Editor” section, and submit your "Accept" recommendation.

Reviewer #1: All comments have been addressed

Reviewer #2: (No Response)

2. Is the manuscript technically sound, and do the data support the conclusions?

Reviewer #1: Partly

Reviewer #2: Partly

3. Has the statistical analysis been performed appropriately and rigorously? 

Reviewer #1: N/A

Reviewer #2: N/A

4. Have the authors made all data underlying the findings in their manuscript fully available?

Reviewer #1: Yes

Reviewer #2: Yes

5. Is the manuscript presented in an intelligible fashion and written in standard English?

Reviewer #1: Yes

Reviewer #2: No

6. Review Comments to the Author

Reviewer #1: Thank you for addressing my comments

Thank you for addressing my comments

Thank you for addressing my comments

Reviewer #2: The manuscript presents only a few images showing possible cytoplasmic localization Pax6 in neuromast hair cells of salamander larvae at one single stage. I am not sure this is a sufficiently novel and complete study to warrant publication. It is necessary to more comprehensively address the appearance of Pax6 in the apical appendages of hair cells.

If no embryos could be obtained from these species for a comprehensive analysis of Pax6 localization in hair cells, the authors may use other amphibians or even fish to perform the study. This will help understand whether the cytoplasmic localization of Pax6 in hair cells is conserved.

7. PLOS authors have the option to publish the peer review history of their article (what does this mean?). If published, this will include your full peer review and any attached files.

Reviewer #1: **Yes: **Jon Martin Collinson

Reviewer #2: No

---

## [Author Response · Author response to Decision Letter 1]

12 Jul 2024

We are grateful for the rapid turnaround of the revised manuscript and appreciate the careful review given by the academic editor and the reviewers. We seek to address reviewer #2’s concerns below.

Point by point response to the reviewers’ comments (copied from e-mail with some omissions and with our responses bulleted)

Reviewer #1: All comments have been addressed.

• Thank you, Dr. Collinson, for your previous comments. We are pleased that you found our responses satisfactory.

Is the manuscript presented in an intelligible fashion and written in standard English?

Reviewer #1: Yes

Reviewer #2: No

• We have carefully read the manuscript, searching for misspellings and grammatical errors, taking advantage of the tools available through Microsoft Word. We identified an error in which we had twice cited a reference in a single sentence (line 247), and we corrected that error. We also changed the wording on the description of hair cells (lines 314-316) to better describe apical appendages in the context of multiple cells. Without further guidance, we are unsure how to address this comment. 

Reviewer #2: The manuscript presents only a few images showing possible cytoplasmic localization Pax6 in neuromast hair cells of salamander larvae at one single stage. I am not sure this is a sufficiently novel and complete study to warrant publication. It is necessary to more comprehensively address the appearance of Pax6 in the apical appendages of hair cells. If no embryos could be obtained from these species for a comprehensive analysis of Pax6 localization in hair cells, the authors may use other amphibians or even fish to perform the study. This will help understand whether the cytoplasmic localization of Pax6 in hair cells is conserved.

• We agree with the reviewer that the manuscript is not extensive; however, PLOS ONE guidelines state that the journal accepts “all rigorous research.” While the web site does not define rigor, a common understanding involves the application of appropriate research methodology and controls to meet stated objectives of the research. Thanks in part to the additional controls proposed by the reviewers, we believe have met the standard for rigor and that we are conservative in our interpretation of the results and conclusions that we draw, specifically that PAX6 localizes to the apical appendages of hair cells of neuromasts, a novel finding that has not been previously reported. Furthermore, although our study system is limited to salamanders of the genus Eurycea, we do show this localization in two different species representing very different ecotypes, indicating some level of conservation during development. Extending the observation to other taxa may or may not reveal conservation; however, the point of this study is simply to report the novel observation of PAX6 localization in the apical appendages of hair cells.

• We also agree that it is necessary to do more to comprehensively address the appearance of PAX6 in hair cells, and we are hopeful that by publishing this paper, we will open the door to other investigators, each with a unique perspective on the implications of this finding and the appropriate follow-up experiments. For our part, we are hoping to pursue the question of whether PAX6 is associated with the cytoskeleton in this system. Our laboratory has no full-time researchers, so we anticipate the study will take at least two years to complete. 

We are grateful for the constructive criticism and the effort expended to review our manuscript. We hope this rebuttal satisfies the concerns raised by the reviewer.

---

## [Decision Letter · Decision Letter 2]

7 Aug 2024

PONE-D-23-32484R2PAX6 protein in neuromasts of the lateral line system of salamanders (*Eurycea*)PLOS ONE

Dear Dr. Garcia,

Thank you for submitting your manuscript to PLOS ONE. After careful consideration, we feel that it has merit but does not fully meet PLOS ONE’s publication criteria as it currently stands. Therefore, we invite you to submit a revised version of the manuscript that addresses the points raised during the review process.

We look forward to receiving your revised manuscript.

Kind regards,

Haitham Abo-Al-Ela, DVM, MSc, PhD

Academic Editor

PLOS ONE

Journal Requirements:

Additional Editor Comments:

Due to several limitations—such as the unavailability of genome sequences, the difficulty in obtaining embryos from the endangered Eurycea, and the inability of the authors to perform additional confirmatory methods—it is crucial to clearly outline these limitations in the manuscript. This information should be included in relevant sections, including the abstract. I strongly recommend using probabilistic language where appropriate when reporting results throughout the manuscript and suggesting further investigatory studies to confirm or refute these findings.

Reviewers' comments:

Reviewer's Responses to Questions

**Comments to the Author**

1. If the authors have adequately addressed your comments raised in a previous round of review and you feel that this manuscript is now acceptable for publication, you may indicate that here to bypass the “Comments to the Author” section, enter your conflict of interest statement in the “Confidential to Editor” section, and submit your "Accept" recommendation.

Reviewer #1: All comments have been addressed

2. Is the manuscript technically sound, and do the data support the conclusions?

Reviewer #1: Yes

3. Has the statistical analysis been performed appropriately and rigorously? 

Reviewer #1: N/A

4. Have the authors made all data underlying the findings in their manuscript fully available?

Reviewer #1: Yes

5. Is the manuscript presented in an intelligible fashion and written in standard English?

Reviewer #1: Yes

6. Review Comments to the Author

Reviewer #1: No further comments.

7. PLOS authors have the option to publish the peer review history of their article (what does this mean?). If published, this will include your full peer review and any attached files.

Reviewer #1: **Yes: **Jon Martin Collinson

---

## [Author Response · Author response to Decision Letter 2]

13 Aug 2024

Point by point response to the academic editor’s and reviewer’s comments (copied from e-mail with some omissions and with our responses bulleted)

Academic Editor: Due to several limitations – such as the unavailability of genome sequences, the difficulty in obtaining embryos from the endangered Eurycea, and the inability of the authors to perform additional confirmatory methods – it is crucial to clearly outline these limitations in the manuscript. This information should be included in relevant sections, including the abstract. I strongly recommend using probabilistic language where appropriate when reporting results throughout the manuscript and suggesting further investigatory studies to confirm or refute these findings.

• The authors acknowledge the limitations of the study and have attempted to address the editor’s comments and previous reviewers’ comments as detailed below in the order items appear in the manuscript (Line numbers are based on the unmarked manuscript):

o To speak probabilistically, in line 18 (Abstract): “…PAX6 … appears to be present in the mechanosensory neuromasts…” and in line 22 (Abstract): “anti-PAX6 antibody labeled structures that were extranuclear….”

o To further communicate the difficulty of obtaining samples, in line 19-22 (Abstract): “…Using immunohistochemistry and confocal microscopy to examine a limited number of larvae of two species, listed by the United States of America’s federal government as threatened (E. nana) or endangered (E. rathbuni)….” 

o To communicate the unavailability of genomic or other sequence data, in line 181-182 (Methods): “Sequence data are not publicly available for salamanders in the genus Eurycea.”

We note that in R2, we were able to add a partial sequence of Pax6 obtained from Eurycea nana and that result is shown in Figure 2.

o To communicate the prevalence and extent of labeling and better flesh out the certitude of our observation (probabilistic language), we reevaluated our images, counted all neuromasts observed and described the labeling in lines 193-201 (Results).

We added to this section information about the number of neuromasts in which labeling was observed in the apical appendages as well as the number of individuals from each of the two species. In short, we did not observe labeling in every neuromast; however, we did observe labeling of the apical appendages in every individual salamander examined.

For R2, we had added in preadsorption experiments for both immunohistochemical and western analyses. When antibodies were preadsorbed in both cases, labeling was reduced below detection, corroborating our inference that the labeling we report reveals the location of PAX6. 

o To further emphasize the limitations of the study, we conclude the Discussion with the paragraph that follows (lines 381 – 390): 

A challenge of extending the work we present here is the availability of specimens of these understudied, threatened and endangered species for which publicly available genomic data is not available. Work is underway to generate transcriptomic data, and preliminary results for the sequence of Pax6 from the transcriptome of Eurycea nana are included herein. We expect to further investigate Pax6 expression in skin, and more precisely in neuromasts in future studies, contingent on the availability of salamander larvae. The immunohistochemical results we report here are nevertheless tantalizing, and though the number of individuals probed with anti-PAX6 antibodies is small, it is worth noting that labeling of the apical appendages of hair cells in the neuromasts was observed in all of them. That it was not observed in all neuromasts could be attributed to a failure to capture the apical appendages in some tissue sections. 

o 

Reviewer #1: All comments have been addressed.

• Thank you, Dr. Collinson, for your previous comments. We are pleased that you found our responses satisfactory.

---

## [Editor Report · Decision Letter 3]

16 Aug 2024

PAX6 protein in neuromasts of the lateral line system of salamanders (*Eurycea*)

PONE-D-23-32484R3

Dear Dr. Garcia,

We’re pleased to inform you that your manuscript has been judged scientifically suitable for publication and will be formally accepted for publication once it meets all outstanding technical requirements.

Kind regards,

Haitham Abo-Al-Ela, DVM, MSc, PhD

Academic Editor

PLOS ONE

---

## [Editor Report · Acceptance letter]

22 Aug 2024

PONE-D-23-32484R3 

PLOS ONE

Dear Dr. García, 

I'm pleased to inform you that your manuscript has been deemed suitable for publication in PLOS ONE. Congratulations! Your manuscript is now being handed over to our production team.

Kind regards, 

on behalf of

Dr. Haitham Abo-Al-Ela 

Academic Editor

PLOS ONE